# Political participation among deaf youth in Great Britain

**Francisco Espinoza**[1,2]*, **Alys Young**[2], **Claire Dodds**[2]

**1** Department of Politics, School of Social Sciences, The University of Manchester, Manchester, United Kingdom, **2** Social Research with Deaf People (SORD), Division of Nursing, Midwifery and Social Work, The University of Manchester, Manchester, United Kingdom

\* francisco.espinoza@manchester.ac.uk

**Data Availability Statement:** A relevant subset from the original data and a script for analysis have been uploaded to a Figshare repository (https://doi.org/10.48420/24968175).

**Funding:** The READY study is funded by the National Deaf Children's Society (UK). The views

## Abstract

Variations in political participation are linked to demographic factors, socioeconomic disparities, and cultural-ethnic diversity. Existing research has primarily explored reduced political involvement among individuals with disabilities, particularly in electoral politics. However, little research has attended the involvement of deaf people specifically. This is of interest because deaf youth are at an intersection of disability, language and cultural identity with their language affiliations and rejection or acceptance of disability evolving through childhood. This study draws from original data collected via an online survey, comprising 163 deaf young respondents aged 16-19 in Great Britain. We compare their levels of political participation with those of general population peers to explore how sociodemographic factors, alongside variations in self-identification as deaf, and meaningful interactions with other deaf people contribute to explain their political engagement. The results challenge conventional wisdom by demonstrating that deaf youth participate more actively in politics than their hearing peers in various forms of political involvement, including collective, contact, and institutional activism. We also recognize differences among deaf youth and propose that social aspects of identity formation, particularly embracing a deaf identity and having deaf friends, can boost certain forms of political engagement. In summary, this study underscores the importance of acknowledging the diversity of deaf youth in terms of affiliation with various forms of deaf identity, rendering their experience different from both disabled and hearing youth. By identifying the factors driving heightened political participation, policymakers and advocates can develop strategies to enhance political engagement among all young people, regardless of their hearing status.

## Introduction

Social mobilization and activism among young people have caught the attention of the media and the general population, including environmental movements such as Extinction Rebellion, Fridays for Future, and School Strike for Climate, and anti-racist movements such as Black Lives Matter (BLM). These activities are part of a broad repertoire of political participation, which is understood as the engagement in activities aimed to influence government actions,

expressed in this article are those of the authors and not necessarily those of NDCS.

**Competing interests:** The authors have declared that no competing interests exist.

either directly impacting policy-making or through the selection of the decision-makers [1]; including electoral turnout and a broader repertoire of non-electoral actions such as joining demonstrations, attending political meetings or boycotting products, among others [2–4]. Despite the attention gained by youth mobilization in the media and recent scholarly research [4–7], little is known about how young people with disabilities engage in these spaces [8], let alone how deaf young people might—the focus of this article.

Deaf young people are an interesting population to consider with respect to forms of political participation because intersectional aspects of their identity extend beyond features such as gender, sexuality, ethnicity and socio-economic status, to more fundamental ontological aspects of what it is "to be deaf" [9]. This is because identity attributions associated with deafness range from passing as hearing, or regarding oneself as a disabled person, to membership of a distinct cultural-linguistic community of users of a signed language based on which full rights of citizenship are claimed [10, 11]. Furthermore, fluidity in identity affiliations associated with being deaf is emerging as a distinct difference in contemporary deaf youth to whom binary distinctions between those who identify with the Deaf Community and those who do not on grounds of signed or spoken language no longer apply [12, 13]. While some deaf activists situate themselves within the disability movement; others prefer to situate within the Deaf rights movement with the affiliation with cultural-linguistic identity often marked by the capitalization of Deaf. This contestation of positionality was neatly summarized in the title of Corker's 1998 volume: "Deaf and disabled or deafness disabled". Intersectional aspects of deaf identity (e.g Black and Deaf, Deaf women, LGBT+ and Deaf) have largely been explored in relation to adults with a strong culturally Deaf identity and who use a signed language with less attention paid to intersectionality in a wider diversity of deaf people and very little attention with regard to deaf youth [14, 15].

In what follows, we consider the applicability of research on the political participation of young disabled people to the experience of deaf youth and review the literature on political participation and signing Deaf people before presenting novel data on the political participation of a cohort of diverse deaf young people between 16 to 19 years old in Great Britain. We define political participation as the active involvement in actions oriented to influence government or policy outcomes, covering engagement in a repertoire of non-electoral activities during the last twelve months [1, 4]. We compare levels and rates of participation amongst deaf young people in the READY cohort [16] with general population youth participation data in the country [17–20]; and then analyze the factors driving deaf young people's political engagement.

Where literature specifically relates to culturally Deaf people we mark it by the use of the capital D; where it does not or there is some ambiguity and/or fluidity of identity attribution we use "deaf" to encompass all. Considering a broad repertoire of non-electoral political participation, our results indicate a statistically significant higher participation of deaf young people in almost every action compared to the general population of similar age, except volunteering where proportions are similar. Among the READY cohort of deaf young people, having deaf friends act as a driver to sign petitions or shared online posts; while identifying as d/Deaf contributes to explaining these activities and also involvement in other collective actions.

## Literature review

The study of political participation among young people in Europe and Britain has evolved over time, from comparing participation levels with older generations—usually on voter turnout—to seeking to understand why and how levels of participation vary amongst young people

[7, 21, 22]. More recently, attention has shifted from the electoral aspects of political participation such as voting to a broader and extended repertoire of political participation and potential driving factors. The change in the optics has permitted a move from condemnation to the recognition of an expanding repertoire of actions, technologies and attitudes toward politics [23–25]. This has led to a greater focus on the various ways in which youth engage with politics, including demonstrations, petitioning, contacting MPs, and other activities beyond electoral turnout; and focus on the differences among young people. Studies have identified, for example, that inequalities derived from social class and educational achievement can explain variation in the levels of political participation among youth [5, 26]. Moreover, young women have been identified as tending to engage more frequently in less confrontational activities, such as petitioning, boycotts, and volunteering; whereas young men tend to participate more frequently in conventional activities such as attending political meetings or contacting politicians [4].

With respect to people with disabilities, research has consistently evidenced a lower level of political participation than among the general population, although largely focusing on election turnout in the US and Europe [27–32]. As previous researchers have mentioned, studies concerning the broader repertoires of political participation among people with disabilities are scarce [33], although people with disabilities report higher levels of participation in political campaigns and higher levels of political interest than in the general population [34]. Additionally, in a Europe-based study, people with disabilities report higher rates of attendance at demonstrations and contact politicians more frequently than those without disabilities [35]. While these findings break with the dominant expectation of participation among people with disabilities; these findings are not clearly differentiated by age cohort. Griffith's study [8] specifically of young people with disabilities focuses on aspects of political participation and activism but only concerning involvement in the UK Disabled People's movement, concluding that young people encounter barriers within the movement itself when perceived to be out of step with older and more established leaders.

Moreover, it is notable that in this range of literature on people with disabilities and political participation, potential differences in motivations, repertoire of activities, and impacts are not differentiated by nature of disability [34, 36, 37]. However, the disabled people's movement itself, whilst largely united in an understanding of the socially produced nature of disability, would also acknowledge differences in lived experiences and challenges between those living with different kinds of disabling conditions. The self-efficacy of the individual and that of the collective in political participation for people with disabilities overlap but are not co-terminus [35]. The homogeneity of the category 'people with disabilities' in current research in political participation thus masks the experience of deaf people who identify or are classified as disabled persons.

In contrast, the literature on Deaf people who are sign language users and political participation is distinguished by core issues of language rights and cultural identity [38]. This is because many countries do not legally recognize sign languages resulting in discrimination on grounds of language use and unequal rights of citizenship [39]. For example, jury service may be denied to sign language users, public bodies may not be required to make their information and services accessible in signed languages. Consequently, research concerning Deaf people and political participation largely focuses on the activism and political campaigning for the recognition of signed languages as fully grammatical, living and indigenous languages of their country of origin [40, 41]. Central to this is the establishing of rights based on language and culture rather than subsumed under disability equality legislation [10, 42–44]. However, this form of political participation for 'legal citizenship' sits alongside literature on what Hall [45]

refers to as 'lived citizenship'. This refers to the everyday activities that embody political participation, which in turn is mediated by language and communication [25].

The barriers to information, communication, and understanding that Deaf people face when the infrastructure of usual political activities are not available through their language impede their ability to take part in the repertoire of political participation in general [25, 46]. Examples might include linguistic access to take part in pressure group meetings, formal information from candidates in elections that is in sign language, participation in surveys, and general political news and information through formal and informal media. The recent "Where is the interpreter?" campaign in response to inaccessible government information during the COVID-19 pandemic in the UK is a case in point [47]. Furthermore, as Emery [48] identified, Deaf community members share a commitment to their own local communities in the mainstream and are seeking involvement in political participation outside of their own concerns as the Deaf community too. Political campaigns and involvement of Deaf youth specifically are also gaining ground although not formally studied. For example, the largely Deaf-youth-initiated campaign by Daniel Jillings to have British Sign Language taught as an examination subject within mainstream schools is a case in point [49].

We posit that some of the specific characteristics of the d/Deaf population may act as potential factors to explain the engagement in political actions. The first is group identity, which has been considered as a driving factor for political participation since the seminal works of Verba and Nie [50]. Group identification does not rely solely on objective conditions but requires three elements: the recognition of membership of a group, a psychological attachment involving value connotations -often the recognition of a disadvantage-, and the willingness to initiate collective actions [51–53]. Researchers have explained how people from minorities and marginalized groups, as well as those with disabilities, can hold the perception of a "linked fate" derived from common experiences such as group-based discrimination or shared circumstances in life. As a consequence, they have more incentives to engage in collective actions to advocate for their needs and improve their position [54–56].

In the case of deaf young people, the notions of linked fate, collective identity, and group action can become more complex. This is because culturally Deaf sign language users would not necessarily see their identity as deriving from their deafness but rather from their shared language. Nonetheless, deafness as a disability is an overlapping category if for no other reason than legislatively those who are not hearing are regarded as disabled [9, 10]. Many deaf people who do not sign would not identify with the Deaf community and many would readily regard themselves as disabled rather than possessing a deaf or Deaf identity. Consequently, collective identity and shared experiences of discrimination and marginalization on the grounds of being deaf are more differentiated. Nonetheless, we would expect that a stronger sense of a shared or collective experience of being d/Deaf should serve as an incentive for political engagement. We refer to this as a d/Deaf identity, which in the context of this article means sharing or having similar life experiences with other deaf or Deaf people, encompassing identifying with others who are d/Deaf and that these experiences, who one feels similar to, may in themselves be shared amongst d/Deaf people(s).

A second potential explanatory factor for the variance in political participation among deaf young people is the extent of interaction with other d/Deaf people, including its diversity and quality. While much of the existing research on social drivers of participation has focused on the role of organizational affiliation [1, 57] or the magnitude of social connections [58, 59], the direct influence of interpersonal connections within the group has been somewhat overlooked. In this context, what seems fundamentally relevant is the impact of meaningful interactions with "politically significant others" [60]. This suggests that even seemingly individual-level

actions, such as wearing a badge or signing a petition, could actually result from interactions with peers who share similar experiences and political inclinations.

In the case of deaf youth, we might expect that their friendships with other deaf individuals can play a crucial role in obtaining information about politics and raising awareness on their shared experiences. Friendship relations with other d/Deaf people are informative of deafness as a collective experience alongside discovering shared motivations for change at the social level through politics. Consequently, we might expect having deaf friends to be a significant driver of their engagement in political actions.

Against the backdrop of limited evidence concerning the political participation of deaf young people, which is distinct from evidence related to young people with disabilities and encompasses both culturally Deaf individuals and those who are not, this study aims to delve into the levels and repertoire of political engagement among deaf youth. To achieve this, we utilized cohort data from The READY Study, a unique longitudinal research project among deaf youth known in England, Scotland, and Wales. We describe the levels of political participation of deaf young people in Great Britain, how these compare to that of the general population, and what factors drive political involvement among deaf youth. We develop and test two explanatory mechanisms to explain variation in political participation among deaf young people: identifying as deaf over audiological categories, and having deaf friends. The overall aims of the work, which were achieved, were:

- To describe the repertoire of political participation among a large cohort of deaf young people;

- To compare the extent and characteristics of political participation among them with those of young people in the general population of the same age;

- To investigate the impact of a d/Deaf identity and meaningful interactions with other d/Deaf people on the variation of levels of political participation among deaf young people.

## Materials and methods

### Data: The READY study

The READY Study (Recording Emerging Adulthood in Deaf Youth) is a prospective, longitudinal cohort study that focuses on deaf young individuals living in England, Scotland, or Wales, who are followed prospectively for up to 5 years [16]. The inclusion criteria were limited to individuals who are permanently deaf -including those with unilateral deafness-, aged between 16 and 19 years at the entry point, and capable of providing informed consent in accordance with the Mental Capacity Act 1998 (England and Wales) and the Incapacity Act 2000 (Scotland), part 5. After screening for eligibility, respondents gave voluntary consent using an online form on REDCap. Deaf young people who were interested in participating could contact the study at age 15, expressing their desire to take part. However, they would not be screened for eligibility and formally consented into the study until they reached the age of 16. Formal approval for the study was granted by the University of Manchester Research Ethics Committee 2 on May 21, 2019. Recruitment of participants began on 25th June 2019, being suspended between March and July 2020 after the first COVID-19 lockdown. After restarting, registration concluded on 30th June 2021.

For this study, we used cross-sectional data from the first round of annual data collection, which included 163 participants [61]. It is important to note that consent forms and all survey data were available in multiple languages and formats to accommodate the diversity of deaf young people, including Written English and Written Welsh, Sign Supported English and Sign

Supported Welsh, and BSL (British Sign Language). In addition, the REDCap data collection and storage platform was adapted to allow participants to switch language/format on a question-by-question basis, enabling them to maximize their comprehension and meet their translanguaging preferences [62]. The self-report survey was conducted online only.

The annual survey covered demographic data, lifestyle data, educational attainment and employment data, as well as specific sections on communication and language, social networks, and political participation. Questions in each section were designed to be comparable to general population cohort study data.

### Research design

#### Hypotheses.

1. Hypothesis 1: Levels of political participation of d/Deaf young people would be higher than those of the general population of the same age.

2. Hypothesis 2: Deaf young people who recognize themselves as having a d/Deaf identity would be more likely to engage in political participation than those who do not.

3. Hypothesis 3: Deaf young people who have meaningful interactions with other d/Deaf people would be more likely to engage in political participation than those who do not.

**Descriptive and comparative analysis.** Our analysis is divided into three parts. In the first, we use descriptive statistics to characterize the levels of political participation among deaf young people. In the READY annual survey, respondents were asked: "During the last year, which of the following have you done?" encompassing eight different possible actions. Participants can reply "I have done this" or "I have not done this". The eight forms of political participation were as follows:

1. Written to or visited your local MP

2. Attended a public meeting about an issue you care about

3. Been on a demonstration or attended a rally

4. Worn or displayed a campaign badge, poster or sticker

5. Bought or boycotted a product for ethical, political or environmental reasons

6. Signed a petition

7. Shared messages on social media about politics or issues of concern

8. Volunteered on a school, college, youth council, or student union

In the case of "volunteering", we have also included responses derived from their occupational status. Respondents were asked, "What are you doing now?" in a multiple response item including a wide spectrum of answers (I am in education, I am in work, I am on an apprenticeship, I am on a training scheme, I am involved in voluntary work, I am not in education or work or training), and those who declare involvement in volunteering were included under the corresponding item.

We compare the answers of respondents in our sample with a benchmark of respondents in the general population. The comparison is made using the European Social Survey 7th, 8th and 9th rounds collected in 2014, 2016 and 2018/19 respectively [18–20], selecting respondents from the United Kingdom in the 15 to 19-year-old group who declare not to be hampered in

their daily activities due to disabilities, long-term illness or mental problems (n = 172). Using exclusively ESS 9th round was dismissed because it only included a small sample of general population of the same age (n = 49), making it difficult to compare with the READY sample. The inclusion of three cross-sectional stages permits an increase in the number of respondents from the same age included in the analysis but, as a trade-off, it would not account for variation associated with the specific years of fieldwork. We also compare data from participants between the ages of 16 to 19 years old who took part in the Community Life Survey (CLS) [17] conducted in England during the years 2019/20 and who do not have a limiting long-term illness or disability (n = 348).

To answer our first hypothesis, we compare the reported means of each activity in the READY sample with that of the general population of the same age in the selected surveys. We conducted a two-sample test of proportions at 95% standard level to assess whether the differences are statistically significant or not. Then, to test our second hypothesis, we use the package MatchIt (v3.4.4) in R to implement an optimal full matching analysis to control for sociodemographic variables that may alter the results due to differences in the composition of the samples and weight them based on the composition. The READY study and ESS samples are matched using standard sociodemographic descriptors associated with political participation among youth in the UK context: ethnicity (1: White; 2: Black, Asian or Minority Ethnic [BAME] in the UK); sex (1: Men, 2: Women); age; and region. Regions were regrouped into Greater London, North of England (including North East, North West, and Yorkshire and the Humber), Midlands (East and West), South and East of England (including South West, South East and East of England), Scotland and Wales. In the case of the CLS study, we have used the subset of respondents living in England (n = 136) and weighted the composition of our sample using sex, ethnicity, region of England, and socioeconomic status. This last variable corresponds to the Index of Multiple Deprivation calculated for each neighborhood or lower-layer super output area (LSOA) (Ministry of Housing, Communities & Local Government, n.d.). We then use generalized linear models with a logistic function (package lme4, v1.1) to assess the magnitude of the difference between READY and respondents from national surveys, controlling for sociodemographic variables and weights.

**Explanatory analysis.** In our third hypothesis, we investigate the factors that promote engagement among deaf young people. Based on previous categorizations of political participation in the UK and Western democracies [2, 3], we have condensed these activities into three different forms of political participation (contact, institutional and collective activism) but adapted their components based on the analysis of internal consistency for each factor and Principal Component Analysis (PCA). Firstly, we have defined "contact activism" as the actions oriented to raise awareness of an issue with a general audience; covering whether respondents have signed a petition or shared messages on social media ($\alpha$ = 0.63). This category includes but is not limited to forms of online activism. Secondly, we have defined "collective activism" as activities involving public commitment to causes in the public space, such as attending a public meeting, participating in demonstrations, wearing a badge, boycotting a product, and volunteering ($\alpha$ = 0.64). Finally, we have defined "institutional activism" as a separate dimension for those respondents who have contacted their representative in Parliament. This variable is composed of only one questionnaire item, as the factor analysis indicates that it does not contribute to the loadings on any of the two factors and its inclusion would reduce the internal consistency of the dimensions. While existing research has treated it as a contact type of political action; the younger age of respondents –many with no legal right to vote yet- and the hierarchical aspect may act as deterrents compared to other forms of contact.

We have operationalized each of these three dimensions as a binary variable measuring whether respondents have engaged with any of the possible items included for that category in

the last twelve months (1 if they have participated; 0 if they have not participated). We use generalized linear models with a binomial function (lme4, v1.1) to assess the contribution of factors associated with deaf identity, meaningful interactions, and sociodemographic indicators.

**Explanatory variables.**   Our model specifications include the usual sociodemographic categories associated with variation in political participation, such as sex, ethnicity, education, and socioeconomic status. The level of qualifications is obtained using the eight-level scale from the Regulated Qualifications Framework (RQF), and socioeconomic status is measured on a 5-point scale using quintiles based on the Index of Multiple Deprivation (IMD). We also include two sociodemographic variables that are particularly relevant to our population. The first is a binary variable indicating whether respondents have additional needs (physical, visual, learning, or mental needs). The second corresponds to their level of deafness, which is operationalized as a five ordinal categories variable following the Consortium for Research in Deaf Education (CRIDE) standard (unilateral, mild, moderate, severe, and profound), including young people with unilateral deafness in the READY study.

In conjunction with the sociodemographic variables, we test for the role of the two relevant mechanisms previously discussed. The first is having a d/Deaf identity (referring to self-identification as d/Deaf, however that may be understood) and the second, having meaningful interactions with other d/D people. We distinguish between them because, along with the differences in the explanatory mechanisms presented in the previous section, as self-identification as d/Deaf does not necessarily imply interactions, of whatever kind, with other d/Deaf people. We operationalize "meaningful interactions" using '1' for those who have d/Deaf friends and '0' if they do not have d/Deaf friends. Deaf friends include online as well as in-person friendships. In the baseline group, 87 respondents (53.4%) reported having a deaf friend, while 76 participants (46.6%) stated that they did not have a deaf friend.

We operationalize "d/Deaf identity" with '1' for respondents who describe themselves as 'deaf' or 'Deaf' and '0' for those who use instead an audiological category associated with a functional definition of disability and hearing loss to describe themselves ("I am disabled", "I have a hearing impairment", "I am hard of hearing", "I am a bit deaf", "I have a hearing loss", "I have a hearing problem"). In referring to identity, we are not making a binary distinction between those who are sign language users and members of the Deaf community and those who are not. Evidence from contemporary deaf youth indicates a more fluid approach to identity recognition and affiliation than in previous times [12, 13]. Consequently, d/Deaf identity might include those who are culturally Deaf but also those who reject a functional disability approach to their identity, seeing themselves as just deaf (whether a sign language user or not). The difference is between those with a form of identity centered on being d/Deaf and those with a form of identity centered on not being fully hearing. In the READY study, 89 participants (54.6%) preferred to identify as "I am deaf", while 74 participants (45.4%) used an audiological category.

## Results and discussion

### Descriptive statistics

We first analyzed the differences between the means for each action of political participation between READY respondents and those in the comparison general population surveys (Table 1). It is notable that in all categories of political engagement, a higher percentage of READY participants declare political involvement in comparison with the CLS and ESS samples. Among READY respondents, 13% declare that they have contacted their MP (institutional activism), which is higher than 5.8% in CLS and 4.7% in ESS.

**Table 1. Political participation among deaf young people and population of their same age.** Reported proportions in the READY study, CLS and ESS.

| Political Action | CLS 2018-19 (N = 348) | ESS 7-9 (N = 172) | READY Study (N = 163) |
|---|---|---|---|
| Contacted their MP | | | |
| Proportion | 20 (5.8%) | 8 (4.7%) | 21 (13.0%) |
| Missing | 3 | 0 | 2 |
| Significance of difference | ** | ** | - |
| Attended meeting or demonstration | | | |
| Proportion | 23 (6.7%) | - | 47 (29.2%) |
| Missing | 3 | - | 2 |
| Significance of difference | *** | - | - |
| Worn a badge or sticker | | | |
| Proportion | - | 15 (8.7%) | 44 (27.2%) |
| Missing | - | 0 | 1 |
| Significance of difference | - | *** | - |
| Attended demonstration | | | |
| Proportion | - | 5 (2.9%) | 20 (12.3%) |
| Missing | - | 0 | 1 |
| Significance of difference | - | ** | - |
| Boycotted a product | | | |
| Proportion | - | 17 (9.9%) | 57 (35.4%) |
| Missing | - | 1 | 2 |
| Significance of difference | - | ** | - |
| Volunteering | | | |
| Proportion | 157 (45.1%) | - | 74 (46.2%) |
| Missing | 0 | - | 3 |
| Significance of difference | Not signif. | - | - |
| Signed a petition | | | |
| Proportion | 93 (27.0%) | 49 (28.5%) | 110 (68.3%) |
| Missing | 3 | 0 | 2 |
| Significance of difference | *** | *** | - |
| Posted online about an issue | | | |
| Proportion | - | 33 (30.6%) | 109 (67.3%) |
| Missing | - | 0 | 1 |
| Significance of difference | - | *** | - |

***$p < 0.001$;

**$p < 0.01$;

*$p < 0.05$;

·$p < 0.1$

In the questions related to contact activism, 68.3% of respondents in READY declare that they have signed a petition—which is more than double the 27% in CLS and 30.6% among ESS respondents. Also, 67.3% have written an online post about an issue they cared about, which is nearly three times the rate of ESS respondents in the UK.

In the items related to collective activism, READY participants also show higher levels of political engagement in almost every item. Among READY respondents, 27.2% wore a badge or a sticker for a campaign they supported, while only 6.1% of CLS respondents did the same; and 35.4% declare that they have boycotted a product, an activity in which only 9.9% of ESS were involved. Regarding participation in public meetings and demonstrations, ESS and CLS

have different wording in their questionnaires. In ESS, only 2.9% declare that they have attended a demonstration, an activity that rises to 12.3% among READY respondents. In the CLS questionnaire, the item includes both attending a demonstration or public meeting, in a single item, so we merged the responses in the READY data. The result shows that 29.2% of READY respondents attended a demonstration or public meeting in contrast to only 6.7% of CLS respondents. The differences between READY respondents and the national benchmark for the population of the same age in CLS and ESS studies are statistically significant in all mentioned items. The only exception is volunteering, which reaches 45.1% among the general population in CLS and 46.2% in the READY study.

## Comparative statistics

Comparing the descriptive statistics of these three samples does not necessarily provide a generalizable description of each group. It could be argued that READY participants may not be representative of all deaf young people of this age as the sample is skewed towards certain demographics. The sample comprises more people from affluent backgrounds and there are considerably more women than men [16]. Thus, the CLS and ESS may represent a broader spectrum of young people in terms of educational attainment and SES than the READY sample. However, this issue is not unique to the READY study as selection biases may result from people with a higher level of interest and educational attainment being more likely to answer a political survey.

To address these differences in the composition of the READY, CLS, and ESS samples, we have used a complementary approach of a matching analysis to compare the magnitude of the difference between the samples while controlling for sociodemographic factors. The READY study and ESS samples were matched using sex, ethnicity, age, qualifications obtained, and UK region of residence. In the case of CLS, we matched the surveys using sex, ethnicity, qualifications obtained, IMD quintiles, and England's region of residence. This way, we can obtain a more accurate comparison through sample weighting and restricting the effect of sociodemographic drivers in uneven samples.

The differences are presented in tables and organized by each dimension of political participation, and then plotted using odds ratios. In Table 2, we show the results for institutional activism (contacting a Member of Parliament MP or Scottish/Welsh country equivalent) with the samples matched with CLS and ESS. After matching the samples using sociodemographic variables, READY respondents are still more likely to have contacted their local representative in Parliament than respondents in CLS but not in the ESS. The statistically significant coefficients for READY respondents indicate that the probability is larger for READY respondents when compared to CLS respondents ($\beta$ = 1.75, SE = 0.44, p < 0.001). In Fig 1, we plot the odds ratio for the comparison between deaf young people and their peers in the general population. In this case, READY respondents are 5.78 times more likely to have contacted their local Parliamentary representative than CLS respondents, but the difference with ESS respondents is not significant.

In the case of contact activism (Table 3), participants in the READY study are more likely to have signed a petition than respondents from the same age in the general population, both in the CLS ($\beta$ = 1.75, SE = 0.24, p <0.001) and ESS samples ($\beta$ = 1.2, SE = 0.29, p < 0.001). READY respondents are also more likely to have written an online post about an issue they care about ($\beta$ = 1.2, SE = 0.32, p < 0.001). As presented in Fig 1, deaf young people are 5.78 times more likely to have signed a petition than respondents in CLS in England and 3.33 times more likely than people of the same age in ESS in Great Britain. Deaf young people are 3.35

**Table 2. Political participation (institutional activism) among READY respondents compared to ESS and CLS data, matched samples.**

| | Contact MP (READY/CLS) | Contact MP (READY/ESS) |
|---|---|---|
| (Intercept) | -2.54 ** | -1.20 |
| | (0.89) | (5.05) |
| Survey: READY | 1.75 *** | 0.73 |
| | (0.44) | (0.52) |
| Ethnicity: White | -0.17 | -1.96 ** |
| | (0.49) | (0.65) |
| Sex: Women | -0.15 | -0.31 |
| | (0.47) | (0.54) |
| Quintile (IMD) | -0.42 ** | |
| | (0.16) | |
| Region: ref. Greater London | | |
| Reg: Midlands | -1.58 | -1.21 |
| | (1.00) | (1.29) |
| Reg: North of England | 0.19 | 1.01 |
| | (0.58) | (0.86) |
| Reg: South and East | -0.24 | -0.11 |
| | (0.61) | (0.93) |
| Reg: Scotland | -1.37 | |
| | (1.30) | |
| Reg: Wales | 0.77 | |
| | (1.15) | |
| Qualifications (RQF level) | 0.28 | 0.60 * |
| | (0.17) | (0.26) |
| Age | -0.08 | |
| | (0.30) | |
| AIC | 166.43 | 135.32 |
| Num. obs. | 436 | 308 |

***$p < 0.001$;

**$p < 0.01$;

*$p < 0.05$;

·$p < 0.1$

times more likely to have written an online post about issues they care about than people of the same age in the general population.

With regard to collective activism (Table 4), READY respondents are more likely to have attended a demonstration in the last year than those in the general population, when compared with ESS respondents ($\beta = 1.34$, SE = 0.52, p = 0.014). In the CLS survey, the question is framed differently as it covers both attending meetings or demonstrations. In this case, differences between READY respondents and CLS in England are also statistically significant ($\beta = 2.07$, SE = 0.32, p < 0.001). After controlling for all other sociodemographic variables, READY respondents are 3.3 more likely to have attended a demonstration than ESS respondents and 12.2 times more likely to have attended either a demonstration or a meeting than CLS respondents, as presented in Fig 2.

In the case of wearing badges or stickers (Table 5), differences between READY and ESS respondents are significant ($\beta = 0.98$, SE = 0.36, p = 0.006), and deaf young people are 2.68

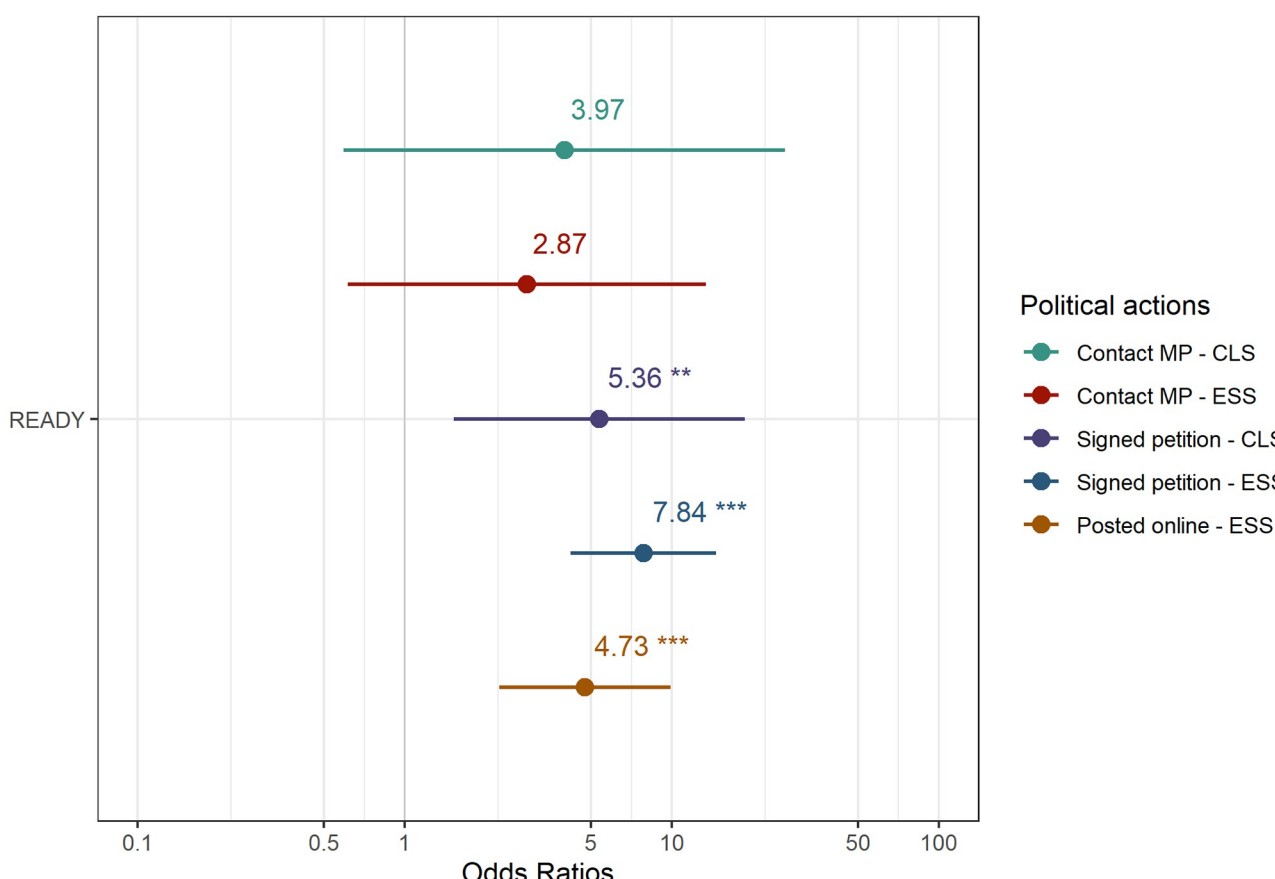

**Fig 1. Differences in the probability of engagement in political participation (odds ratio).** The figure presents the odds ratio of involvement for READY participants in signing petitions, posting online and contacting their MP, compared to the probabilities among the general population.

times more likely to have been involved in this activism. Differences are also significant for engaging in boycotting a product ($\beta = 1.45$, SE = 0.35, $p < 0.001$), and deaf young people are 4.27 times more likely to have boycotted a product compared to ESS. Finally, the analysis of volunteering shows that there are no significant differences between deaf young people and their hearing peers in England.

## Explanatory factors for variation among deaf young people

In the final part of our analysis, we aim to explain differences in the variation of involvement among deaf young people in contact activism, institutional activism, and collective activism. As presented in Tables 6–8, we use two different model specifications for each of them, contrasting the contribution of sociodemographic factors only (Sociodemographic only: sex, age, ethnicity, degree of deafness, qualifications, socioeconomic status, and having additional needs) with two particular factors as previously defined (full): having meaningful interactions with other deaf people (i.e., having a deaf friend) and d/Deaf identity. We can compare the Akaike Information Criterion (AIC) at the bottom of the table to assess the performance of the sociodemographic models with those, including the d/Deaf identity and friendship factors for each repertoire of political participation. This indicator penalizes overfitting, which is optimal when more variables are included, and the standard interpretation is that models with

**Table 3. Political participation (contact activism) among READY respondents compared to ESS and CLS data, matched samples.**

| | Signed a petition (READY/CLS) | Signed a petition (READY/ESS) | Posted online (READY/ESS) |
|---|---|---|---|
| (Intercept) | -2.37*** | 7.03** | 3.56 |
| | (0.47) | (2.55) | (2.91) |
| Survey: READY | 1.75*** | 1.20*** | 1.21*** |
| | (0.24) | (0.29) | (0.33) |
| Ethnicity: White | 0.60* | -1.24* | -0.67 |
| | (0.28) | (0.51) | (0.58) |
| Sex: Women | 0.47. | 0.68* | 0.51 |
| | (0.25) | (0.29) | (0.33) |
| Quintile (IMD) | 0.07 | | |
| | (0.08) | | |
| Region: ref. Greater London | | | |
| Reg: Midlands | 0.70. | 0.14 | 1.09. |
| | (0.37) | (0.54) | (0.58) |
| Reg: North of England | 0.10 | -0.06 | 0.97. |
| | (0.35) | (0.51) | (0.52) |
| Reg: South and East | 0.48 | -0.38 | -0.36 |
| | (0.33) | (0.50) | (0.51) |
| Reg: Scotland | | -1.64** | -1.60* |
| | | (0.62) | (0.70) |
| Region: Wales | | -2.77** | -1.63* |
| | | (0.93) | (0.80) |
| Qualifications (RQF) | 0.03 | 0.83*** | 0.49** |
| | (0.09) | (0.16) | (0.16) |
| Age | | -0.49** | -0.30. |
| | | (0.15) | (0.17) |
| AIC | 514.33 | 339.69 | 285.19 |
| Num. obs. | 435 | 307 | |

***$p < 0.001$;

**$p < 0.01$;

*$p < 0.05$;

.$p < 0.1$

minimum AIC are preferred over those with larger AIC. We can also use Pseudo R-squared Nagelkerke (R2N) as a metric to assess and contrast the proportion of the variance explained by our models. This is an adaptation from the regular R2 used in ordinal least squares models and, in a similar manner to that indicator, larger values are preferred as they provide a better explanation of the results.

In the case of contact activism, the Model 2 (Table 6) including d/Deaf identity and meaningful interactions explains more variance without overfitting (R2N 0.31, AIC 119.9), resulting in an improvement from the sociodemographic Model 1 (R2N 0.18, AIC 128.9). In both models, the differences between men and women are significant, considering the inclusion of all other explanatory variables. Deaf women are more likely to write online posts and sign petitions than their deaf male peers ($\beta = 1.03$, SE = 0.52, p = 0.048). The difference is robust and still significant after we include the variables related to d/Deaf identity and interactions ($\beta = 1.39$, SE = 0.57, p = 0.015). None of the other sociodemographic variables seem to indicate

**Table 4. Political participation (collective activism) among READY respondents compared to ESS and CLS data, matched samples.**

| | Demonstration (READY/ESS) | Demonstration or meeting (READY/CLS) |
|---|---|---|
| (Intercept) | 7.13 | -3.80*** |
| | (4.90) | (0.75) |
| Survey: READY | 1.34* | 2.50*** |
| | (0.55) | (0.37) |
| Ethnicity: White | 0.10 | -0.08 |
| | (1.07) | (0.41) |
| Sex: Women | 1.19* | -0.32 |
| | (0.59) | (0.38) |
| Age | -0.68* | |
| | (0.30) | |
| Region: ref. Greater London | | |
| Reg: Midlands | 0.45 | -0.53 |
| | (0.90) | (0.54) |
| Reg: North of England | -0.18 | 0.10 |
| | (0.90) | (0.46) |
| Reg: Scotland | -2.11 | |
| | (1.65) | |
| Reg: South and East | -1.40 | -1.37** |
| | (1.03) | (0.53) |
| Reg: Wales | -1.38 | |
| | (1.33) | |
| Qualifications (RQF level) | 0.35 | 0.17 |
| | (0.27) | (0.14) |
| Quintile | | 0.22 |
| | | (0.14) |
| AIC | 134.51 | 239.57 |
| Num. obs. | 308 | 436 |

***$p < 0.001$;

**$p < 0.01$;

*$p < 0.05$;

$p < 0.1$

differences based on sociodemographic factors between READY respondents. The inclusion of friendship and identity variables results of statistical significance as, in the case of people who have d/Deaf friends, they are more likely to be involved in these practices ($\beta = 1.60$, SE = 0.61, p = 0.008). In the case of people who identify as d/Deaf over those whose identity encompasses a measure of not being fully hearing, the difference reaches the standard threshold of statistical significance ($\alpha = 0.05$) by a narrow margin ($\beta = 1.29$, SE = 0.64, p = 0.045).

The average marginal effect (AME) indicates that, after controlling for all other relevant sociodemographic characteristics, respondents who have deaf friends are 18.3 percentage points more likely to have written an online post or signing a petition than those who do not have a deaf friend. In the case of identity, respondents who identify themselves as deaf are 15 percentage points more likely to have engaged in these forms of activism.

Two other sociodemographic factors that indicate potential differences in this repertoire are the IMD quintile, as respondents living in less deprived areas tend to participate more ($\beta =$

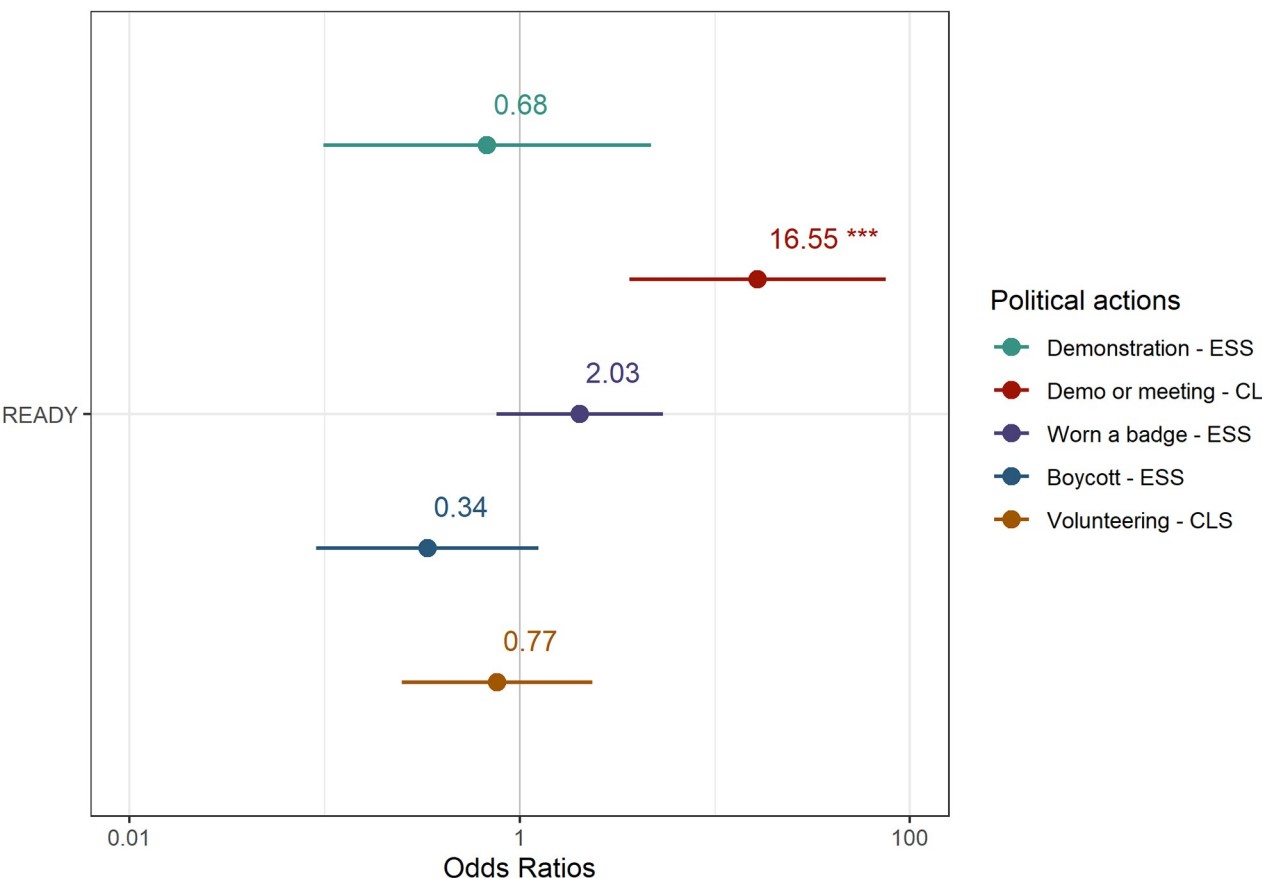

**Fig 2. Differences in the probability of engagement in political participation (odds ratio).** The figure presents the odds ratio of involvement for READY participants in attending demonstrations, political meetings, wearing a badge, boycotting a product and volunteering, compared to the probabilities among the general population.

0.35, SE = 0.20, p = 0.08), and those who have additional needs could be less likely to engage ($\beta$ = -0.98, SE = 0.55, p = 0.07), but they do not reach statistical significant under the conventional threshold ($\alpha$ = 0.05).

In the case of institutional activism (Table 7), the inclusion of d/D identity and meaningful interactions generates a slight increase in the performance from R2N 0.09 in Model 3 to 0.14 in Model 4, while maintaining a similar AIC rate (from 121.3 to 121.6). The sociodemographic specification (Model 3) results in a significant effect of the degree of hearing loss and engagement with institutional practices ($\beta$ = 0.43, SE = 0.22, p = 0.051), if a more flexible threshold is assumed ($\alpha$ = 0.1). Contrary to expectations from dominant literature on disabilities and participation, individuals with profound deafness could be more likely to have contacted their MP/local parliamentary representative in the past year, but this effect becomes insignificant when controlling for variables related to community membership (i.e. contact with other d/Deaf people). It seems that individuals with greater degrees of deafness are more inclined to participate in institutional activism, but this effect may potentially be influenced by the social environment that fosters their activism. This could imply that having deaf friends and identifying as d/Deaf are capturing some of the variance that the level of hearing loss was explaining in the initial model, turning the effect of hearing loss to work through these additional variables.

**Table 5. Political participation (collective activism) among READY respondents compared to ESS and CLS data, matched samples.**

|  | Worn a badge (READY/ESS) | Boycott (READY/ESS) | Volunteer (READY/CLS) |
|---|---|---|---|
| (Intercept) | 8.67* | 0.15 | -0.76. |
|  | (3.44) | (3.18) | (0.41) |
| Survey: READY | 0.99** | 1.45*** | 0.11 |
|  | (0.36) | (0.35) | (0.22) |
| Ethnicity: White | -1.12. | -0.87 | 0.20 |
|  | (0.61) | (0.62) | (0.25) |
| Sex: Women | 0.83* | 0.57 | -0.14 |
|  | (0.38) | (0.35) | (0.22) |
| Age | -0.66** | -0.20 |  |
|  | (0.21) | (0.19) |  |
| Region: ref. Greater London |  |  |  |
| Reg: Midlands | 0.59 | 1.25. | 1.17*** |
|  | (0.70) | (0.72) | (0.34) |
| Reg: North of England | -0.00 | 0.38 | 0.61. |
|  | (0.69) | (0.71) | (0.32) |
| Reg: Scotland | -1.26 | -0.35 |  |
|  | (0.97) | (0.83) |  |
| Reg: South and East | -0.25 | 0.51 | 0.28 |
|  | (0.68) | (0.70) | (0.31) |
| Reg: Wales | -0.40 | -0.53 |  |
|  | (0.89) | (0.98) |  |
| Qualifications (RQF level) | 0.44* | 0.52** | -0.16. |
|  | (0.19) | (0.17) | (0.08) |
| Quintile |  |  | 0.09 |
|  |  |  | (0.07) |
| AIC | 230.90 | 255.29 | 576.34 |
| Num. obs. | 308 | 306 | 438 |

Ultimately, after controlling for community membership variables, no statistically significant predictors emerge, indicating that neither sociodemographic nor community membership-related factors can explain levels of institutional activism.

In the case of collective activism (Table 8), the inclusion of d/Deaf identity and meaningful interactions slightly improves the proportion of variance explained compared to (R2N 0.04 in Model 5 to 0.11 in Model 6) with an improvement on the model performance as AIC (176.9 to 173.7). Overall, the inclusion of d/Deaf identity and meaningful interactions increases the performance of the models without overfitting, which is the desired outcome. Nonetheless, the model specifications do not provide an even explanation for all forms of political participation, as it explains a larger proportion of contact activism than other forms of political engagement. The sociodemographic variables in the model account for a small fraction of the variance (0.04), and none of the factors reach statistical significance at the conventional threshold ($\alpha = 0.05$). The effect of IMD quintile ($\beta = 0.24$, SE = 0.14, p = 0.09) could be considered as significant under a more flexible threshold ($\alpha = 0.1$) and has a positive coefficient, indicating a potential direct association between more participation among respondents from less deprived areas.

When including community membership variables in the model, d/Deaf identity has a significant effect, aligning with our expectation that respondents who identify as d/deaf are more

**Table 6. Contact activism among deaf young people, sociodemographic and deaf-related factors.**

|  | Model 1 (Soc. only) | Model 2 (full) |
|---|---|---|
| (Intercept) | −4.46 | −1.21 |
|  | (4.75) | (5.40) |
| Sex: Women | 1.03* | 1.39* |
|  | (0.52) | (0.57) |
| Age | 0.33 | 0.17 |
|  | (0.30) | (0.33) |
| Ethnicity: White | 0.85 | 0.48 |
|  | (0.55) | (0.63) |
| Level of hearing loss (5 CRIDE cat.) | −0.17 | −0.65* |
|  | (0.19) | (0.27) |
| Qualifications (RQF) | −0.03 | −0.00 |
|  | (0.20) | (0.22) |
| Quintile (IMD) | 0.22 | 0.35. |
|  | (0.18) | (0.20) |
| Have additional needs | −0.74 | −0.98. |
|  | (0.50) | (0.55) |
| Having deaf friends |  | 1.60** |
|  |  | (0.61) |
| d/Deaf identity |  | 1.29* |
|  |  | (0.64) |
| AIC | 128.98 | 119.92 |
| R2N | 0.18 | 0.31 |
| Num. obs. | 135 | 135 |

***$p < 0.001$;

**$p < 0.01$;

*$p < 0.05$;

.$p < 0.1$

likely to be involved in politically organized collective activities ($\beta$ = 1.0, SE = 0.47, p = 0.03). In general, respondents who identify as deaf are 19.5 percentage points more likely to have engaged in collective activism in the last 12 months than those who identify with other categories. However, having deaf friends has no statistically significant effect on collective activities ($\beta$ = 0.63, SE = 0.42, p = 0.13).

## Discussion

In relation to our first hypothesis, we have provided evidence that deaf young people engage in higher levels of political participation than the general population of the same age in the UK. More deaf young people have engaged in contact activism than those in the general population, through signing petitions (CLS, ESS) and posting online (ESS). READY participants are also more likely to engage in collective activism by attending meetings or demonstrations (CLS), attending demonstrations (ESS), wearing badges (ESS) and boycotting (ESS). The levels of volunteering seen do not follow the same trend (CLS); however, involvement in volunteering amongst the READY cohort is no lower than in the general population either. Results regarding contacting a politician are equivocal. Comparison to CLS indicates that deaf people in READY are more likely to contact a politician, but there is no difference when compared to

**Table 7. Institutional activism among deaf young people, sociodemographic and deaf-related factors.**

|  | Model 3 (Soc. only) | Model 4 (full) |
|---|---|---|
| (Intercept) | −4.64 | −2.56 |
|  | (4.73) | (4.88) |
| Sex: Women | −0.35 | −0.32 |
|  | (0.61) | (0.63) |
| Age | 0.04 | −0.09 |
|  | (0.29) | (0.30) |
| Ethnicity: White | 0.59 | 0.34 |
|  | (0.65) | (0.71) |
| Level of hearing loss (5 CRIDE cat.) | 0.43˙ | 0.21 |
|  | (0.22) | (0.25) |
| Qualifications (RQF) | 0.32 | 0.33 |
|  | (0.26) | (0.26) |
| Quintile (IMD) | −0.19 | −0.15 |
|  | (0.19) | (0.19) |
| Have additional needs | 0.09 | 0.03 |
|  | (0.53) | (0.55) |
| Having deaf friends |  | 0.90 |
|  |  | (0.60) |
| d/Deaf identity |  | 0.67 |
|  |  | (0.65) |
| AIC | 121.31 | 121.60 |
| R2N | 0.09 | 0.14 |
| Num. obs. | 134 | 134 |

***$p < 0.001$;

**$p < 0.01$;

*$p < 0.05$;

˙$p < 0.1$

those in ESS. Along with the comparatively lower rates of engagement in contacting a MP, a potential explanation is the larger time-period covered by the three cross-sectional applications of ESS (2014, 2016, 2019), potentially conflating with variation across time.

Concerning our second hypothesis, we have provided partial evidence to state that the effect of identifying as d/Deaf (whether part of the culturally Deaf community or deaf as distinguished from disabled/hearing impaired) fosters political participation. It is significant in explaining collective and contact political practices; but not institutional activism. While the effect of group identity on political participation has been largely proposed as an explanatory factor for political participation [50, 51], we provide empirical evidence on how identity becomes a variable of distinction among deaf young people in Britain.

Our third hypothesis explored whether the effect of meaningful interactions with other d/Deaf people (i.e, having a deaf friend) boosts participation. This is proven about contact activism (i.e., signing petitions or posting online) but not in collective or institutional activism. Having deaf friends shows relevant differences among deaf young people, suggesting that contact with other deaf people can boost engagement in a pool of political activities. Neither d/Deaf identity nor meaningful interactions with other d/Deaf people increase the probability of institutional activism (i.e., contacting their MP).

**Table 8. Collective activism among deaf young people, sociodemographic and deaf-related factors.**

|  | Model 5 (Soc. only) | Model 6 (full) |
| --- | --- | --- |
| (Intercept) | 1.67 | 4.22 |
|  | (3.49) | (3.79) |
| Sex: Women | 0.23 | 0.31 |
|  | (0.45) | (0.47) |
| Age | −0.11 | −0.26 |
|  | (0.21) | (0.23) |
| Ethnicity: White | −0.56 | −1.02. |
|  | (0.50) | (0.56) |
| Level of hearing loss (5 CRIDE cat.) | 0.00 | −0.24 |
|  | (0.14) | (0.17) |
| Qualifications (RQF) | 0.06 | 0.09 |
|  | (0.17) | (0.17) |
| Quintile (IMD) | 0.25. | 0.30. |
|  | (0.15) | (0.16) |
| Have additional needs | 0.24 | 0.24 |
|  | (0.41) | (0.43) |
| Having deaf friends |  | 0.62 |
|  |  | (0.43) |
| d/Deaf identity |  | 1.00* |
|  |  | (0.47) |
| AIC | 176.96 | 173.76 |
| R2N | 0.04 | 0.11 |
| Num. obs | 134 | 134 |

***$p < 0.001$;

**$p < 0.01$;

*$p < 0.05$;

.$p < 0.1$

As such, our results indicate levels of variation among deaf young people in terms of our hypothesis: identifying as a deaf person and having meaningful interactions with other deaf people. In addition, part of the variation also falls in line with previous empirical evidence describing sociodemographic variation. For instance, deaf women are more likely than men to sign petitions or post online, a gender gap previously described for these and other less confrontational activities in the general population in Europe [4]. Interestingly, we also identified how people with deeper hearing loss may engage in contacting their MP, but the effect does not sustain after including variables associated with identity and connections. We do not believe the causal mechanism relies on the level of deafness itself; instead, the inclusion of group identity and meaningful contacts suggest that there is a fostering effect associated with these factors.

## Conclusion

We present novel data that fills a substantial gap in the literature by examining the levels of political participation among deaf young people and how they compare to the general population. In line with empirical research studying political participation among youth, our definition of political participation encompasses a wide range of activities [4]. Our findings are

encouraging, as deaf young people participate more in political activities than their hearing peers of the same age in the country. To ensure comparability with other survey studies in Great Britain, we controlled for sociodemographic and regional differences factors. Additionally, we found that individuals who hold meaningful interactions with other d/Deaf people in friendship relationships are more likely to participate in "contact activism" -writing an online post, signing a petition, and those who identify as d/Deaf are more likely to engage in collective actions.

Our results challenge the traditional expectations of lower levels of political participation among marginalized populations. While research has shown lower levels of electoral turnout among people with disabilities [27–32]; we build on research that has expanded the repertoire of actions [34] to demonstrate larger rates than the general population. Moreover, in line with empirical research providing an explanatory mechanism around collective experiences [35], we provide evidence of significant factors to explain variation among deaf young people in Great Britain. In particular, we provide partial evidence that group identity and having meaningful interactions with other deaf people are factors contributing to engaging in political activities. Therefore, having deaf friends and identifying as deaf -instead of audiological or disabilities categories- can provide stronger incentives for people to participate in certain types of political activities.

Our research makes notable contributions to two distinct academic debates. Firstly, we address an empirical gap by examining, comparing and explaining political participation among deaf youth. Our work underscores the significance of sensitive distinctions in political studies, particularly for a group often subsumed under overarching categories such as 'people with disabilities' and within a generation frequently characterized as disinterested in politics. The outcomes of our study distance from conventional expectations, showing the heightened engagement of deaf youth in a diverse range of political activities. Secondly, we engage in discussion with deaf studies by studying the involvement of deaf youth with the public sphere and explaining its potential drivers. The examination of the role of the social dimensions of deafness and political participation demonstrates the diversity among deaf people, contributing to a deeper understanding of the diverse experiences of deaf people in political life and serving as valuable information for activists and deaf organizations.

However, our study has limitations, as is typical of observational and self-reported data. Although we recruited a relatively large number of participants compared to similar studies, we do not intend to generalize our conclusions beyond our sample. Moreover, it is challenging to disentangle differences between age, period, and cohort using cross-sectional data, as seen in research on youth participation. Further research is necessary to explore the causal pathways, and how changes in identification and meaningful interactions with other deaf young people can increase political participation.

## Acknowledgments

The full READY research team also includes Dr Garry Squires, Dr Katherine Rogers and Dr Helen Chilton who are co-investigators on this research project. We would like to thank our anonymous reviewers in this journal, our reviewers at the MPSA Annual Conference 2023 and Dr Cristián Iturriaga for their comments on previous versions of this manuscript. We sincerely thank all deaf young people who took the time to respond to our survey and provide data on their lives.

## Author Contributions

**Conceptualization:** Francisco Espinoza, Alys Young, Claire Dodds.

**Data curation:** Francisco Espinoza.

**Formal analysis:** Francisco Espinoza.

**Funding acquisition:** Alys Young.

**Investigation:** Francisco Espinoza, Alys Young, Claire Dodds.

**Methodology:** Francisco Espinoza, Alys Young, Claire Dodds.

**Project administration:** Francisco Espinoza, Claire Dodds.

**Supervision:** Alys Young.

**Validation:** Claire Dodds.

**Visualization:** Francisco Espinoza.

**Writing – original draft:** Francisco Espinoza, Alys Young, Claire Dodds.

**Writing – review & editing:** Francisco Espinoza, Alys Young, Claire Dodds.

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
