## [Decision Letter · Decision Letter 0]

21 Dec 2023

PONE-D-23-35756Political participation among deaf youth in Great BritainPLOS ONE

Dear Dr. Espinoza,

Thank you for submitting your manuscript to PLOS ONE. After careful consideration, we feel that it has merit but does not fully meet PLOS ONE’s publication criteria as it currently stands. Therefore, we invite you to submit a revised version of the manuscript that addresses the points raised during the review process.

We look forward to receiving your revised manuscript.

Kind regards,

Nabeel Al-Yateem, PhD

Academic Editor

PLOS ONE

Journal Requirements:

The READY study is funded by the National Deaf Children’s Society (UK). The views expressed in this article are those of the authors and not necessarily those of NDCS.

The READY study is funded by the National Deaf Children’s Society (UK). The views expressed in this article are those of the authors and not necessarily those of NDCS. The full READY research team also includes Dr Garry Squires, Dr Katherine Rogers and Dr Helen Chilton who are co-investigators on this research project. We sincerely thank all deaf young people who took the time to respond to our survey and provide data on their lives. 

The READY study is funded by the National Deaf Children’s Society (UK). The views expressed in this article are those of the authors and not necessarily those of NDCS.

5. We note that you have indicated that there are restrictions to data sharing for this study. For studies involving human research participant data or other sensitive data, we encourage authors to share de-identified or anonymized data. However, when data cannot be publicly shared for ethical reasons, we allow authors to make their data sets available upon request. For information on unacceptable data access restrictions, please see http://journals.plos.org/plosone/s/data-availability#loc-unacceptable-data-access-restrictions. 

Reviewers' comments:

Reviewer's Responses to Questions

**Comments to the Author**

1. Is the manuscript technically sound, and do the data support the conclusions?

Reviewer #1: Partly

Reviewer #2: Yes

2. Has the statistical analysis been performed appropriately and rigorously? 

Reviewer #1: Yes

Reviewer #2: Yes

3. Have the authors made all data underlying the findings in their manuscript fully available?

Reviewer #1: No

Reviewer #2: Yes

4. Is the manuscript presented in an intelligible fashion and written in standard English?

Reviewer #1: Yes

Reviewer #2: Yes

5. Review Comments to the Author

Reviewer #1: Topic

The article is developed around a very pertinent topic for different research fields as sociology, youth studies, education and inclusion: political participation of deaf young people. Through a quantitative approach the study aims to compare not deaf and deaf young people levels of political participation. Less attention has been given to policital participation among young people from the deaf culture.

Abstract:

The abstract includes information regarding the research gap, method, results and recommendations, but would benefit from a clear theoretical and conceptual standpoint underlying the research and data analysis.

Introduction: the introductions starts by providing an analytical context for youth participation, in particular recent forms of participation or involvement in social movements and the absence ok knowledge regarding how young deaf are part of those.

Although the concept of interseccionality is mentioned it seems that deafness is the main characteristic (variable) that interfere in being involved in practices of participation and it sounds that the author is pointing to a more homogenous perspective of Deaf community.

Moreover, the concept of political participation needed to be defined and clarified at this point of the article as in the end of the introduction some results and claims are presented but without any clear guidance regarding the main conceptual organisers.

Lit. Review

This section focus on contribution of the literature to understand political participation of people with disabilities and raise relevant questions regarding levels of participation among that population and discusses specifically how participation among deaf people. However, I suggest to further discuss some aspects that sounds incomplete. For example, the sentence in page 3 sound incomplete, our at least needed some additional information: “In contrast, the literature on Deaf people who are sign language users and political 75 participation is distinct”. Could you elaborate a bit more on this idea.

This section needs some review at two levels: level of structure as it is a bit circular and moving back and forward around the issue of disables and Deaf youth, mixing political rights and citizenship discussion with identity deriving sign language. Although it is understandable the reference to all of this subjects the section needs a clear sequence of topics to which are given relevance by the main aim of this article. This leads to the second level: content level. To include literature review regarding political participation of young people in general sounds necessary, not only because implicitly it is against those theoretical perspectives that Deaf youth participation is understood, but also because this study aims to compare Deaf youth with youth in general. The debate on political participation and what it is included in that definition seems necessary in this context.

Suggestion: I suggest to include the last paragraph related to READY either in the introduction or in the Methods section.

Method

Methodological procedures, as well as specifically Ethical procedures are clear and presented in detail. Procedures regarding how hypothesis were approached are detailed and well explained, supported by similar actions taken before. Regarding the third hypothesis, the PCA resulted in three factors with consistency and with alpha values that are good for social sciences. One factor has only one item and this would deserve a clear justification. Moreover, theoretical support of the 3 factors would be beneficial for clarity and pertinence.

Results and discussion

The descriptive statistics could provide information to understand results intra sample (READY respondents) to understand if deafness is the main variable to explain different results. This aspect is more clear in the following sections presenting interesting results, but the gender differences found, for example, are not clearly explained and discussed. This section is mainly presented at the descriptive level. I suggest proposing an explanatory framework to support results and to connect those with theory. There are signs that the Deaf young people are diverse intra sample and this is not completely explored.

---

What is the contribution of this study to advance new layers of understanding regarding youth participation?

In the abstract the last sentence points to some contributions to policies development, but the article either in results or in the conclusion does not elaborate on this.

Reviewer #2: An excellent manuscript to be published which compares the levels of political participation among deaf youth with those of their peers in the general population and discusses how sociodemographic factors, self-identification as deaf, and meaningful interactions with other deaf people contribute to their political engagement. However, this manuscript needs more discussion, which can be improved by referring to the highlighted literature review.

6. PLOS authors have the option to publish the peer review history of their article (what does this mean?). If published, this will include your full peer review and any attached files.

Reviewer #1: No

Reviewer #2: No

---

## [Author Response · Author response to Decision Letter 0]

15 Jan 2024

Dear Dr Al-Yateem, 

Academic Editor PLOS ONE

Thank you for your letter and the opportunity to submit a revised version of our manuscript. We also appreciate the reviewers’ comments, which will surely help us to enhance the manuscript's clarity. 

Please find our point-by-point response to them and the journal requirements below.

Section 1. Journal Requirements:

RESPONSE: We have ensured that the style and file naming conventions have complied with PLOS ONE requirements. We have ensured that we are using the most updated PLOS ONE template for LaTeX (v3.6, August 2022), and thus our file should match with the journal standards.

RESPONSE: Our main dataset is already a repository already in accordance with our funder’s requirements and it is under embargo, but we have created a smaller anonymized version with the purpose of replication of the analysis developed in this research article. The data is available in Figshare and can be accessed using following the DOI 10.48420/24968175

The READY study is funded by the National Deaf Children’s Society (UK). The views expressed in this article are those of the authors and not necessarily those of NDCS.

RESPONSE: We can confirm the following: "The funders had no role in study design, data collection and analysis, decision to publish, or preparation of the manuscript." We have included this amended Role of Funder statement in the cover letter. 

The READY study is funded by the National Deaf Children’s Society (UK). The views expressed in this article are those of the authors and not necessarily those of NDCS. The full READY research team also includes Dr Garry Squires, Dr Katherine Rogers and Dr Helen Chilton who are co-investigators on this research project. We sincerely thank all deaf young people who took the time to respond to our survey and provide data on their lives. 

The READY study is funded by the National Deaf Children’s Society (UK). The views expressed in this article are those of the authors and not necessarily those of NDCS.

RESPONSE: We have removed the funding statement in the acknowledgements section and as per point 3 above changed it in line with journal requirements in the online submission form, and the cover letter as explained in Point 3. We have retained the acknowledgement of the other members of the READY study team in the acknowledgement section.

5. We note that you have indicated that there are restrictions to data sharing for this study. For studies involving human research participant data or other sensitive data, we encourage authors to share de-identified or anonymized data. However, when data cannot be publicly shared for ethical reasons, we allow authors to make their data sets available upon request. For information on unacceptable data access restrictions, please see http://journals.plos.org/plosone/s/data-availability#loc-unacceptable-data-access-restrictions. 

RESPONSE: There are no ethical or legal restrictions on sharing the de-identified data set. Following the restrictions associated with the time embargo on the whole data set, we have created a specific version for replication analysis. The data set and the script syntax used in the analysis were uploaded to the public depository Figshare, and can be accessed using the DOI 10.48420/24968175. We have updated the Data Availability statement in the submission form to reflect this.

Section 2. Comments from reviewer 1

Topic: The article is developed around a very pertinent topic for different research fields as sociology, youth studies, education and inclusion: political participation of deaf young people. Through a quantitative approach the study aims to compare not deaf and deaf young people levels of political participation. Less attention has been given to policital participation among young people from the deaf culture.

Abstract: The abstract includes information regarding the research gap, method, results and recommendations, but would benefit from a clear theoretical and conceptual standpoint underlying the research and data analysis.

RESPONSE: The reviewer is correct that we did not treat young people from deaf culture as a distinct group because we were problematizing the extent of affiliation that this generation of young deaf people had with deaf culture, rather than perceiving it as a binary distinction. We have strengthened the abstract to clearly show the conceptualisation of deaf youth as neither disabled nor culturally deaf, but that the fluidity of their affiliations in terms of identity and contact with other deaf people was itself of potential influence in the diversity of their political participation and in relation to our disability and general population comparators.

Introduction: the introductions starts by providing an analytical context for youth participation, in particular recent forms of participation or involvement in social movements and the absence ok knowledge regarding how young deaf are part of those.

Although the concept of interseccionality is mentioned it seems that deafness is the main characteristic (variable) that interfere in being involved in practices of participation and it sounds that the author is pointing to a more homogenous perspective of Deaf community.

Moreover, the concept of political participation needed to be defined and clarified at this point of the article as in the end of the introduction some results and claims are presented but without any clear guidance regarding the main conceptual organisers.

RESPONSE: It was not our intention to present a homogeneous view of the deaf community or deaf people more broadly – in fact, the opposite. Therefore, it is helpful that the reviewer seems to suggest we have done this to enable us to clarify further. We have done so in the text and with further references. We have defined and clarified the concept of political participation we are using here in the introduction. 

Literature Review: This section focus on contribution of the literature to understand political participation of people with disabilities and raise relevant questions regarding levels of participation among that population and discusses specifically how participation among deaf people. However, I suggest to further discuss some aspects that sounds incomplete. For example, the sentence in page 3 sound incomplete, our at least needed some additional information: “In contrast, the literature on Deaf people who are sign language users and political 75 participation is distinct”. Could you elaborate a bit more on this idea.

RESPONSE: We have changed the paragraphing to show more clearly the relationship between our statement on the distinct nature of deaf cultural political participation campaigns and the examples given in what had previously been the next paragraph. We have also added some further clarifying text.

[Literature review] This section needs some review at two levels: level of structure as it is a bit circular and moving back and forward around the issue of disables and Deaf youth, mixing political rights and citizenship discussion with identity deriving sign language. Although it is understandable the reference to all of this subjects the section needs a clear sequence of topics to which are given relevance by the main aim of this article. This leads to the second level: content level. To include literature review regarding political participation of young people in general sounds necessary, not only because implicitly it is against those theoretical perspectives that Deaf youth participation is understood, but also because this study aims to compare Deaf youth with youth in general. The debate on political participation and what it is included in that definition seems necessary in this context. Suggestion: I suggest to include the last paragraph related to READY either in the introduction or in the Methods section.

RESPONSE: We have previously incorporated information on political participation among youth in the general population within the literature review section. However, we recognize the need for a more explicit explanation of the specific elements included in the data related to political participation for the clarity of this article. To address this, we have reorganized part of the literature review. We have also enhanced the transitions between these various bodies of literature, aiming to establish more explicit connections between them.

We note that the reviewer has used the capitalised D in reference to their comments on Deaf youth. However, we have been clear in our distinction between deaf youth in general and only using the capitalised form when relating clearly and firmly to those who are members of Deaf culture and claim a Deaf identity based on sign language and culture. This is distinct from other forms of deaf identity that may not be on the grounds of sign language and Deaf culture, which we discuss in the paragraph beginning ‘group identity’. Therefore, we would suggest that some of the issues the reviewer raises perhaps arise from that difference in their point of view with our usage of the terms deaf youth and Deaf youth. Nonetheless, we have reiterated through some additional material that the sign language arguments concerning culturally Deaf people are synonymous with claims to citizenship and political rights as these are denied through lack of recognition of the status of signed languages and legal rights in many countries. 

About moving the last paragraph related to READY to the methods section, we followed the journal guidelines that stated that the research questions should follow on at the end of the literature review section and to make those clear we required this background paragraph concerning the READY study. We pick up on further details of that study later on in the methods section. We would therefore prefer to keep this paragraph where it is currently located in the literature review.

Methods: Methodological procedures, as well as specifically Ethical procedures are clear and presented in detail. Procedures regarding how hypothesis were approached are detailed and well explained, supported by similar actions taken before. Regarding the third hypothesis, the PCA resulted in three factors with consistency and with alpha values that are good for social sciences. One factor has only one item and this would deserve a clear justification. Moreover, theoretical support of the 3 factors would be beneficial for clarity and pertinence.

RESPONSE: We have added clarity in this section. Our definitions of the dimensions are based on previous definitions by empirical research, which we have adapted following the analysis of internal consistency. 

Results and discussion: The descriptive statistics could provide information to understand results intra sample (READY respondents) to understand if deafness is the main variable to explain different results. This aspect is more clear in the following sections presenting interesting results, but the gender differences found, for example, are not clearly explained and discussed. This section is mainly presented at the descriptive level. I suggest proposing an explanatory framework to support results and to connect those with theory. There are signs that the Deaf young people are diverse intra sample and this is not completely explored.

RESPONSE: The discussion section has been enhanced by linking our results back with the main theoretical expectations and existing bibliography. We have provided further comments on the gender differences and connected them with previously cited research covering differences in the general population. We have strengthened the role of the explanatory hypothesis presented and linked back to the theoretical explanations presented. 

What is the contribution of this study to advance new layers of understanding regarding youth participation? In the abstract the last sentence points to some contributions to policies development, but the article either in results or in the conclusion does not elaborate on this.

RESPONSE: We have highlighted the contribution of our research in the discussion and then in the conclusion sections. We have made sure to revisit the theoretical debates presented in the literature review section to provide a clearer connection with our findings. We have also made explicit our contributions in the conclusion section.

Section 3. Comments Reviewer 2

An excellent manuscript to be published which compares the levels of political participation among deaf youth with those of their peers in the general population and discusses how sociodemographic factors, self-identification as deaf, and meaningful interactions with other deaf people contribute to their political engagement. However, this manuscript needs more discussion, which can be improved by referring to the highlighted literature review.

RESPONSE: We have augmented the engagement with previously cited literature, including now more explicit references in the discussion section and the conclusions of the manuscript. In the first, we connect our hypothesis back to the main theory and to the empirical findings of previous research. In the conclusions, we have also debated more broadly on the implications of the paper, the connections with previous findings and specific contributions of the article.

We extend our sincere appreciation to the reviewers for generously dedicating their time and expertise to evaluating our article. We are genuinely grateful for the insightful comments provided. After thorough consideration, we have implemented the suggested modifications and believe that we have met the requested criteria.

We look forward to receiving updates.

---

## [Decision Letter · Decision Letter 1]

11 Mar 2024

Political participation among deaf youth in Great Britain

PONE-D-23-35756R1

Dear Dr. Espinoza,

We’re pleased to inform you that your manuscript has been judged scientifically suitable for publication and will be formally accepted for publication once it meets all outstanding technical requirements.

Kind regards,

Nabeel Al-Yateem, PhD

Academic Editor

PLOS ONE

Additional Editor Comments (optional):

Reviewers' comments:

Reviewer's Responses to Questions

**Comments to the Author**

1. If the authors have adequately addressed your comments raised in a previous round of review and you feel that this manuscript is now acceptable for publication, you may indicate that here to bypass the “Comments to the Author” section, enter your conflict of interest statement in the “Confidential to Editor” section, and submit your "Accept" recommendation.

Reviewer #2: All comments have been addressed

Reviewer #3: All comments have been addressed

2. Is the manuscript technically sound, and do the data support the conclusions?

Reviewer #2: Yes

Reviewer #3: Yes

3. Has the statistical analysis been performed appropriately and rigorously? 

Reviewer #2: Yes

Reviewer #3: Yes

4. Have the authors made all data underlying the findings in their manuscript fully available?

Reviewer #2: Yes

Reviewer #3: Yes

5. Is the manuscript presented in an intelligible fashion and written in standard English?

Reviewer #2: Yes

Reviewer #3: Yes

6. Review Comments to the Author

Reviewer #2: The manuscripts underscore the importance of acknowledging the diversity of deaf youth in terms of affiliation with various forms of deaf identity, rendering their experience different from both disabled and hearing youth. All the comments have been addressed and the manuscript is recommended for publication.

Reviewer #3: "Political participation among deaf youth in Great Britain" is a very-well written paper which has an important contribution to make to the literature on political participation by a special focus on deaf youth in Great Britain. As a reviewer who is reading the paper in its second round, I also had the possibility to read the review of Reviewer 1 and the responses of the authors. As far as I can see, all the comments/suggestions of the Reviewer 1 was taken seriously and the paper in its actual form addresses all these points. The paper does not only present novel research and discusses thouroughly the findings of this novel research. These findings challenge the existent literature discussing the reduced political involvements among individuals with disabilities as they demonstrate that deaf youth participate more actively in politics than their hearing peers. The paper also demonstrates that deaf youth is also not homogeneous and underlines the complexity and intersectionality involved with this category. The conclusion successfully summarizes the main findings and also highlight the limitations of the study for further research. In conclusion, I believe that he paper meets all the criteria of the journal and will make an important contribution to the literature on political participation.

7. PLOS authors have the option to publish the peer review history of their article (what does this mean?). If published, this will include your full peer review and any attached files.

Reviewer #2: No

Reviewer #3: No

---

## [Editor Report · Acceptance letter]

22 Mar 2024

PONE-D-23-35756R1 

PLOS ONE

Dear Dr. Espinoza, 

I'm pleased to inform you that your manuscript has been deemed suitable for publication in PLOS ONE. Congratulations! Your manuscript is now being handed over to our production team.

Kind regards, 

on behalf of

Dr. Nabeel Al-Yateem 

Academic Editor

PLOS ONE